# Phase-Dependent Response to Electrical Stimulation of Cortical Networks during Recurrent Epileptiform Short Discharge Generation In Vitro

**DOI:** 10.3390/ijms25158287

**Published:** 2024-07-29

**Authors:** Anton V. Chizhov, Vasilii S. Tiselko, Tatyana Yu. Postnikova, Aleksey V. Zaitsev

**Affiliations:** 1Centre Inria d’Universite Cote d’Azur, 06902 Valbonne, France; 2Computational Physics Laboratory, Ioffe Institute, Saint Petersburg 194021, Russia; 3Laboratory of Complex Networks, Center for Neurophysics and Neuromorphic Technologies, Moscow 121205, Russia; vasily.tiselko@gmail.com; 4Laboratory of Molecular Mechanisms of Neural Interactions, Sechenov Institute of Evolutionary Physiology and Biochemistry of the Russian Academy of Sciences, Saint Petersburg 194223, Russia; tapost2@mail.ru (T.Y.P.); aleksey_zaitsev@mail.ru (A.V.Z.)

**Keywords:** epilepsy, mathematical model, closed-loop stimulation, sensitivity function, interdischarge interval distribution, refractory density model

## Abstract

The closed-loop control of pathological brain activity is a challenging task. In this study, we investigated the sensitivity of continuous epileptiform short discharge generation to electrical stimulation applied at different phases between the discharges using an in vitro 4-AP-based model of epilepsy in rat hippocampal slices. As a measure of stimulation effectiveness, we introduced a sensitivity function, which we then measured in experiments and analyzed with different biophysical and abstract mathematical models, namely, (i) the two-order subsystem of our previous Epileptor-2 model, describing short discharge generation governed by synaptic resource dynamics; (ii) a similar model governed by shunting conductance dynamics (Epileptor-2B); (iii) the stochastic leaky integrate-and-fire (LIF)-like model applied for the network; (iv) the LIF model with potassium M-channels (LIF+KM), belonging to Class II of excitability; and (v) the Epileptor-2B model with after-spike depolarization. A semi-analytic method was proposed for calculating the interspike interval (ISI) distribution and the sensitivity function in LIF and LIF+KM models, which provided parametric analysis. Sensitivity was found to increase with phase for all models except the last one. The Epileptor-2B model is favored over other models for subthreshold oscillations in the presence of large noise, based on the comparison of ISI statistics and sensitivity functions with experimental data. This study also emphasizes the stochastic nature of epileptiform discharge generation and the greater effectiveness of closed-loop stimulation in later phases of ISIs.

## 1. Introduction

Studying the effects of network stimulation on pathological repetitive discharge generation can aid in understanding the mechanisms of generation and the principles of their control. Varying the timing of stimulation between discharges may affect its effectiveness. The phase-response curve (PRC) analysis was developed to study the effects of phase-dependent stimulation [1,2]. This method has been widely used in neuroscience to characterize the activity of neurons, to distinguish their types, and to analyze synchronization. Examples of the PRC analysis can be found in the paper collection by Schultheiss et al. [3]. However, the PRC is only applicable in cases where the stimulation leads to small perturbations of the limit cycle of the considered dynamical system, i.e., where the stimuli shift but do not reset the phase of the oscillations. In contrast, we are considering a case where weak stimuli can, with some probability, induce new discharges and reset the phase. As far as we know, no conventional analysis tool has been developed for this case. Therefore, we propose a new approach based on measuring interspike interval (ISI) distributions and evaluating phase-dependent sensitivity to stimulation.

The primary objective of our study is to determine which of the mathematical models better describes the recurrent epileptiform activity of network activity that resembles status epilepticus [4]. Is it an oscillator that receives noise and generates irregular discharges, or a purely stochastic oscillator that remains silent in the absence of noise? Is there a specific biophysical mechanism, such as synaptic resource depletion or shunting, that relates to inhibition leading to discharge termination? To answer this question, we consider several models listed in Table 1. Note that we are considering only threshold models, which are more realistic in the case of epileptiform activity than well-studied phase oscillators with weak interactions [5,6,7]. Depending on the parameters, the models considered in our study generate irregular discharges either in the supra-threshold regime, where the limit cycle is perturbed by noise (referred to as the ‘oscillator’ regime) or in the subthreshold regime (referred to as the ‘stochastic oscillator’ regime).

The leaky integrate-and-fire (LIF) model is the simplest model considered. It is applied here not to a single neuron but to the network in which epileptiform bursts resemble spikes. The LIF model describes the generation of bursts as a renewal process.

Our basic biophysical model is then obtained as a reduction in the previously proposed model named ‘Epileptor-2’ [8]. The full original model replicates both ictal and interictal epileptic discharges by considering neuronal membrane polarization, synaptic resource dynamics, and ionic dynamics. Ictal discharges, which last for a long time, are influenced by ionic dynamics, while interictal-like short discharges (SDs) occur when ionic gradients are disturbed but remain quasi-stationary and can be modeled by assuming constant ionic concentrations. The mathematical model of SDs is represented by a simple, second-order system of ordinary differential equations (ODEs). This model accurately replicates SDs as large-amplitude stochastic oscillations, which are observed as bursts of spikes in a single neuron. This system is referred to as Model 1. According to Model 1, each SD is terminated by a decrease in synaptic activity that occurs during discharge due to depletion of the pool of ready-to-release glutamate-containing vesicles.

Alternatively, in Model 2, the termination of SD is governed by shunting. Shunting gradually increases during the discharge due to the opening of relatively slow channels, such as sodium- and calcium-dependent potassium channels, together with the cation-permeable channels of after-spike depolarization.

For the mentioned models, we compare the discharge generation statistics and the response to stimulation. The effect of stimulation is characterized by the function of sensitivity to short stimuli applied at different phases of activity between discharges.

## 2. Results

### 2.1. Experiments: Regime of Short Discharge Generation and Weak Stimulation

This study utilized a 4-aminopyridine in vitro model to induce repetitive interictal activity in cortico-hippocampal slices of rat brains (Figure 1). In this model, interictal activity originates in the hippocampal subfield CA3 and propagates through the hippocampal–entorhinal loop [9]. The mean ISI is typically slightly longer than 1 s and can vary between experimental sessions, and can also slowly vary during a single recording. In quasi-stationary regimes, the mean ISI remains constant. The ISIs fluctuate around the mean on the timescale of the quasi-stationary regimes, and their statistics are characterized by the ISI distribution (Figure 1C).

The aim of this study is to investigate the mechanisms of discharge generation in the quasi-stationary regime, as well as the nature of stochasticity and oscillations. Additionally, we investigate the efficacy of a possible control of the discharge generation using weak stimulation. To achieve this, we have chosen conventional stimulation using an extracellular electrode inserted in Schaffer collaterals. In order to probe the mechanisms of discharge generation, we applied short and weak pulses of stimulating current at different phases between the discharges. The probability of evoked population responses to stimulation depends on the phase of stimulation, as well as the strength and duration of the current. In this study, we observed the stochastic occurrence of population responses depending on the phase, while the stimulation current strength was chosen to be subthreshold for each individual slice at a fixed minimal duration of the current step. The subthreshold stimulation current elicited artificial discharges with a non-zero probability, but not for every pulse. The delays of each evoked discharge after the stimulus were negligible compared to the ISIs.

To eliminate slow trends in discharge frequency, we analyzed the efficiency of stimulation by examining the ratios of the ISI with stimulation to the previous control ISI for each pair of successively occurring discharges, $T_{stim}/T$ (Figure 1B), rather than the ISIs themselves. We skipped the interval following each pair of the control interval and the interval with stimulation to allow for all the perturbations of ionic gradients and the other phase-resetting network effects of stimulation to relax. The distribution of ISI ratios shows an initial peak-shaped component that corresponds to the evoked population responses occurring just after the stimuli (Figure 1D). The amplitude of this component is dependent on the phase. To characterize the phase-dependency of the stimulation efficiency, a sensitivity function is introduced below as the ratio of the artificially evoked and natural components of the distribution.

### 2.2. Experiments: Phase-Dependent Sensitivity to Stimulation

To characterize the effect of stimulation dependence on the stimulation phase, we estimate the fraction of discharges successfully evoked. The analysis is as follows:

An i-indexed stimulus is applied at the interval followed by a control interval of the duration $T_{i}$, which contributes to the distribution of control intervals $P_{con}(T)$. For a given phase of stimulation, the time moment of stimulation is ${\varphi \;T}_{i}$. If a discharge appears earlier than ${\varphi \;T}_{i}$, then such discharge is treated as a miss, the corresponding interval is not accounted for, and a new control interval is considered. If a discharge appears after the time moment ${\varphi \;T}_{i}$, then the corresponding interval $T_{stim,i}$ contributes to the distribution of the intervals with stimulation $P_{stim}(T_{stim})$ and the distribution of the ratio of the intervals with and without stimulation $P_{z}(T_{stim}/T)$ (Figure 1D).

In $P_{z}(z),$ one can distinguish the evoked and natural components (Figure 1E). The first peak-shaped component is followed by the stimulus and can be roughly approximated by the shape $\alpha \left(z\right)=\left(\frac{z}{{\delta z}^{2}}\right)\exp\left(-\frac{z}{\delta z}\right)$ for z>0 and 0 otherwise. For a given φ, the second component can be predicted from $P_{con}(T)$ as the distribution $P_{X/Y}^{\varphi }(X/Y)$ of the ratio X/Y of values X and Y, where X and Y are distributed according to $P_{stim}$ and $P_{con}$, respectively; the ratio distribution is zeroed at the interval of X/Y from 0 to φ, and re-normalized. Hence, it is calculated as
(1)$$P_{X/Y}^{\varphi }(z)=\left\{\begin{array}{c} 0,\;for\;z\leq \varphi ;\;\\ P_{X/Y}(z)\;/\int_{0}^{\varphi }P_{X/Y}(z)\;dz \end{array}\right\}$$
where
(2)$$P_{X/Y}(z)=\int_{-\infty }^{\infty }\left|y\right|\;P_{stim}\left(zy\right)\;P_{con}\left(y\right)\;dy$$

Hence, the total distribution is
(3)$$P_{z}\left(z\right)=\gamma \;\alpha \left(z-\varphi \right)+(1-\gamma )P_{X/Y}^{\varphi }\left(z\right)$$
where γ is the measure of the contribution of evoked responses to the statistics. This measure characterizes the efficiency of stimulation.

We evaluated the stimulation efficiency γ in our experiments by fitting data with Equation (1) (Figure 1E). Throughout the series of experiments, we observed an increase in the stimulation efficiency with an increase in the stimulation phase φ (Figure 1F).

### 2.3. LIF Model: Sample Traces

In the simplest of our models, we assume that the discharge generation is a threshold process (Figure 2A), where some potential V(t) relaxes to a zero level with a time constant τ, undergoes noise, and resets to $V_{reset}$ each time when it crosses a threshold $V^{T}$. These processes constitute the LIF-like model (see Section 4). According to the same protocol as in the experiment, the stimulation is applied at a phase φ, i.e., at t=φ T, where T is the control interval (denoted also as $T^{con}$ in figures). Some of the stimulations result in spikes (Figure 2A, arrows), and the others do not. Even for such a simple model as LIF, the calculation of the sensitivity function through the ISI distributions and its parameter dependence analyses are computationally expensive. That is why we further apply a semi-analytical method of ISI calculation.

### 2.4. An Analytical Approximation of the ISI Distribution with the Refractory Density Approach

The calculation of ISI distributions is a computationally expensive procedure, which prevents an expansive analysis of their dependence on parameters. However, for stochastic threshold models, the calculation of ISI distributions, $P_{con}$ and $P_{stim}$, can be performed with the help of the refractory density (RD) approach [10,11].

#### 2.4.1. RD Approach for Arbitrary Time-Dependent Process and Arbitrary Threshold-Type Network Model

Here, for the consideration of a stochastic network, we propose to use the approach that was elaborated for a stochastic neuron or a population of similar uncoupled neurons each receiving the same input but individual noise. In the probability density approach, the evolution of such a population is described in the phase space consisting of the state variables of a single neuron [12,13]. Such consideration of a population is equivalent to the probabilistic consideration of a single stochastic neuron, in which a neuron is found in a certain state (and consequently, a certain position in the phase space) with some probability density ρ. In the particular case called the refractory density approach [14], the neuronal (or network) states are distributed in the space of a single independent variable—the time elapsed since the last spike (or discharge), $t^{*}$. The probability density ρ is called the refractory density.

For arbitrary time-dependent processes, the density $\rho \left(t,t^{*}\right)$ is governed by the equation
(4)$$\frac{\partial \rho }{\partial t}+\frac{\partial \rho }{\partial t^{*}}=-\rho H\left({t,t}^{*}\right),$$
where $H(t,t^{*})$ is the hazard that evaluates the probability for a neuron (or a network) to fire at the state $t^{*}.$ The firing rate ν(t) and the boundary condition for ρ(t,0) follow from normalization $\int_{+0}^{\infty }\rho \left(t,t^{*}\right)dt^{*}=1$. The condition applied to Equation (4) gives
(5)$$\nu (t)\equiv \rho (t,0)=\int_{+0}^{\infty }\rho (t,t^{*})H({t,t}^{*})dt^{*}$$

As shown in [10], for a wide range of models, including LIF and reduced to threshold-type HH neurons that receive a white Gaussian noise, the hazard function can be quite accurately approximated as a function of the mean potential $U(t,t^{*})$ and its total time derivative dU/dt. (For a threshold neuron model, the potential and gating variables are governed by the transport equations obtained after the substitution of $d/dt=\partial /\partial t+\partial /\partial t^{*}$.) The approximation for H is as follows:(6)$$H(t,t^{*})=A+B,$$
where
$$A=\frac{g_{tot}}{C}e^{0.0061-1.12\theta -0.257\theta^{2}-0.072\theta^{3}-0.0117\theta^{4}},$$
$$B=\sqrt{2}{\left[-\frac{d\theta }{dt}\right]}_{+}\sqrt{\frac{2}{\pi }}\frac{\exp(\theta^{2})}{1+\mathrm{erf}\left(\theta \right)},$$
$$\theta =\frac{V^{T}-U(t,t^{*})}{\sqrt{2}\sigma_{V}}\sqrt{\frac{g_{tot}(t,t^{*})}{g_{L}}},$$
$${\left[x\right]}_{+}=\{0,if\;x<0;1\;othewise\}$$

Thus, the hazard function H depends on the potential $U(t,t^{*})$, its total time derivative dU/dt, the total membrane conductance $g_{tot}$ and parameterized by constant capacitance C, the threshold $V^{T}$, and the dispersion of voltage fluctuations $\sigma_{V}$ that is expressed through the dispersion of white current noise $\sigma_{I}$ as $\sigma_{V}=\sigma_{I}/(g_{L}\sqrt{2})$.

We further demonstrate how the parameters of ISI distribution can be found if the hazard function H is known.

#### 2.4.2. RD Approach for Steady States

In steady states, we obtain from Equation (4)
(7)$$\frac{d\rho }{dt^{*}}=-\rho H\left(t^{*}\right)$$

After the integration of Equation (7) with the boundary condition $\rho \left(t^{*}=0\right)=\nu$, we obtain
(8)$$\rho \left(t^{*}\right)=\nu \exp\left\{-\int_{0}^{t^{*}}H\left(t^{\prime }\right){dt}^{\prime }\right\}$$
and the firing rate ν is obtained from the conservation law $\int_{0}^{\infty }\rho \left(t^{*}\right)\;dt^{*}=1$, i.e.,
(9)$$\nu ={\left[\int_{0}^{\infty }\exp\left\{-\int_{0}^{t^{\prime }}H\left(t^{*}\right)dt^{*}\right\}{dt}^{\prime }\right]}^{-1}$$

The ISI distribution is proportional to the flux ρH because the time since the last spike $t^{*}$ is equal to T with respect to the ISI distribution, i.e.,
(10)$$P(T)=\frac{\rho \left(T\right)H(T)}{\nu }$$

The mean ISI is
(11)$$<T>\equiv \int_{0}^{\infty }TP\left(T\right)dT/\int_{0}^{\infty }P\left(T\right)dT=\frac{1}{\nu }\int_{0}^{\infty }t^{*}\rho H\left(t^{*}\right)dt^{*}$$

#### 2.4.3. The RD Approach for a Single Interspike Interval

If the stimulus is weak, then the threshold might not be reached. This case is non-stationary, even if the membrane potential distribution in $t^{*}$ is constant in time. The RD approach allows us to evaluate the probability of the next discharge generation if the present discharge is observed, $P_{next}$. By setting $\rho \left(t,0\right)=const$, even in non-stationary cases, we return to Equation (7) because the boundary condition for Equation (4) and the source term H are independent of time. The integration of Equation (7) gives $\rho \left(t^{*}\right)=\rho \left(0\right)\;\exp\left\{-\int_{0}^{t^{*}}H\left(t^{\prime }\right)\;{dt}^{\prime }\right\}$, and we obtain $P_{next}$ as the ratio of the total flux to the initial amount $\rho \left(0\right)$, i.e., $P_{next}=\int_{0}^{\infty }\rho \left(t^{*}\right)\;H\left(t^{*}\right)\;dt^{*}/\rho \left(0\right)$. Substituting $\rho \left(t^{*}\right)$, we obtain $P_{next}=\int_{0}^{\infty }\exp\left\{-\int_{0}^{t^{*}}H(t^{\prime })\;{dt}^{\prime }\right\}\;H\left(t^{*}\right)\;dt^{*}$.

Therefore, the next discharge occurs with the probability $P_{next}$; and the ISI distribution is determined by Equation (10), which can be rewritten as
(12)$$P\left(T\right)\;=H(T)\exp\left\{-\int_{0}^{T}H\left(t^{\prime }\right){dt}^{\prime }\right\}$$

This formula expresses the ISI distribution via the hazard function.

The derived formulas allow us to efficiently calculate the parameters of the ISI distribution if the behavior of the potential U, governed by a certain model of a neuron or network, is known.

#### 2.4.4. LIF Model: Analytical Approximation of ISI Distribution

For the LIF neuron determined by Equations (27) and (28) of the Methods section, the mathematical expectation of the membrane potential in a steady state, i.e., $U(t^{*})$, is described by the following equation:(13)$$C\frac{dU}{dt^{*}}=-g_{L}U+I,\;U\left(0\right)=V_{reset}$$

In the case without a stimulus, the control case with $U\left(t^{*}\right)=U_{con}\left(t^{*}\right)$, the integration gives
(14)$$U_{con}\left(t^{*}\right)=V_{reset}+\left(\frac{I}{g_{L}}-V_{reset}\right)\left(1-e^{-t^{*}g_{L}/C}\right)$$

In the case of additional stepwise stimulation with the current pulse of the amplitude $I_{stim}$, the duration ${∆t}_{stim}$ and the onset time ${t=t}_{stim}$, the disturbed potential is
(15)$$U\left(t^{*}\right)=U_{con}\left(t^{*}\right)+{∆U}_{stim}\left(t^{*}\right),$$
where
(16)$$\begin{array}{c} {∆U}_{stim}\left(t^{*}\right)=\theta \left(t^{*}-t_{stim}\right)\theta \left(t_{stim}+{∆t}_{stim}-t^{*}\right)\frac{I_{stim}}{g_{L}}\left(1-e^{-\frac{t^{*}g_{L}}{C}}\right)\\ +\theta \left(t^{*}-t_{stim}-{∆t}_{stim}\right)\frac{I_{stim}}{g_{L}}\left(1-e^{-{∆t}_{stim}g_{L}/C}\right) \end{array}$$

Therefore, P(T) can be calculated with Equation (10), where $\rho \left(t^{*}\right)$ is given by Equation (8), $H\left(t^{*}\right)$ is from Equation (6), ν is from (9), and $U\left(t^{*}\right)$ and $\frac{dU}{dt^{*}}$ are from (15) and its derivative, where, in turn, $U_{con}\left(t^{*}\right)$ is from (14) and ${∆U}_{stim}\left(t^{*}\right)$ is from (16). 

The steps of calculation of the ISI distribution, $P\left(T\right)$, are illustrated in Figure 2B. In the control case without stimulation (black curves in Figure 2B), the membrane potential U exponentially evolves from the reset level $V_{reset}$ to 0. When U is about the threshold $V^{T}$, the pulse of the hazard H emerges, and the density ρ drops accordingly. The middle of this dropping phase approximately corresponds to the most probable time moment of spiking and, consequently, to the mean ISI in the control case, <T>, which is denoted here as $T_{0}$. The obtained distribution $P_{con}\left(T\right)$ is shown in Figure 2B, next to the bottom panel, in the black line. 

In the case of stimulation (red curves in Figure 2B), a small spike of membrane potential emerges, which leads to the sharp spike of H, the small but sudden drop of ρ, and the sharp peak of $P_{stim}\left(T\right)$ (Figure 2B, next to the bottom panel, red line). The rest of the distribution after the peak corresponds to the unprovoked discharges. 

Further, we will also use the distributions cut at the interval from 0 to the moment $t^{*}=\varphi \;T_{0}$:(17)$$P_{con}^{\varphi }(T)=\left\{\begin{array}{c} 0,\;for\;T\leq \varphi T_{0};\;\\ P_{con}(T)\;/\int_{0}^{\varphi T_{0}}P_{con}(z)dz \end{array}\right\},\;P_{stim}^{\varphi }(T)=\left\{\begin{array}{c} 0,\;for\;T\leq \varphi T_{0};\;\\ P_{stim}(T)\;/\int_{0}^{\varphi T_{0}}P_{stim}(z)dz \end{array}\right\}$$

The distribution $P_{stim}^{\varphi }$ as a function of the ratio of ISI to the mean control interval $T_{0}$ is plotted in Figure 2B, at the bottom, with a red line.

The correctness of the ISI distribution with the RD approach is justified by comparison with the numerical simulations in Figure 3A. As seen, the direct numerical solution of the stochastic LIF model converges to the RD solution in both cases with and without stimulation (compare gray to black and rose to red, respectively).

#### 2.4.5. LIF Model: Numerical and Analytical Solutions for Two Types of Stimulation

Because a stimulus in our LIF model reflects not aconventional current injection into a neuron but instead an extracellular stimulus, here, we pose the question of whether the stimulus does affect the potential after the end of the stimulus or not. The two variants are expressed by the presence or absence of the last term in Equation (16). In these two cases, the ISI distributions are shown in Figure 3A, where $P_{stim.\;traced}(T)$ corresponds to the presence of that term and vice versa for $P_{stim}(T)$. A comparison of the shapes of these distributions (green and red in Figure 3A) with the experimental ones (Figure 1D) justifying in favor of the latter case (green) is shown. Indeed, whereas the shape of the evoked component significantly depends on the phase of stimulation, the shape of the resting component does not. Therefore, in further analysis of the LIF-based ISI distributions, we consider the case without the last term in Equation (16).

#### 2.4.6. LIF Model: Phase-Dependent Sensitivity to Stimulation

For the sake of simplicity of analytical consideration, in the LIF model, we fix the time of stimulation $t_{stim}$, thus assuming it to be independent of the previous interdischarge interval. In these conditions, we evaluate the contribution of evoked responses into the statistics of intervals, defined by Equation (3), as the integral difference of the distributions with and without stimulus in the interval of stimulation $t_{stim}<T<t_{stim}+{∆t}_{stim}$:(18)$$\gamma =\int_{t_{stim}}^{t_{stim}+{∆t}_{stim}}(P_{stim}^{\varphi }\left(T\right)-P_{con}^{\varphi }\left(T\right))dT/\left(1-\int_{t_{stim}}^{t_{stim}+{∆t}_{stim}}P_{con}^{\varphi }\left(T\right)dT\right)$$

For the sake of visualization, we approximate the distribution $P_{stim}^{\varphi }\left(T\right)$ with $\tilde{P_{stim}^{\varphi }}\left(T\right)$ as follows:(19)$$\tilde{P_{stim}^{\varphi }}\left(T\right)\approx T_{0}\left(\gamma \alpha \left(T-t_{stim}\right)+\left(1-\gamma \right)P_{con}^{\varphi }\left(T\right)\right)$$
with $\alpha \left(z\right)=\left(\frac{z}{{\delta z}^{2}}\right)\exp\left(-\frac{z}{\delta z}\right)$ and $\delta z=\gamma exp(-1){\left(\underset{t_{\mathrm{stim}}<T<t_{\mathrm{stim}}+{∆t}_{\mathrm{stim}}}{\max}P_{stim}^{\varphi }\left(T\right)\right)}^{-1}$.

The function $\tilde{P_{stim}^{\varphi }}\left(T\right)$ well reproduces the amplitude of the pulse evoked by the stimulus and the shape of the rest of $P_{stim}^{\varphi }\left(T\right)$ that corresponds to unprovoked discharges (Figure 2B, bottom panel, green line). It justifies γ as an appropriate measure of sensitivity. 

#### 2.4.7. LIF Model: Dependence of ISI Distribution and Sensitivity Function on Parameters 

The RD method for the ISI distribution calculation allows us to analyze the dependence on parameters. As seen in Figure 2C (top panel), the elevation of the threshold $V^{T}$ smoothes the ISI distribution $P_{con}\left(T\right)$ and increases the mean ISI $T_{0}$; the sensitivity function flattens (Figure 2D, top). A negative threshold $V^{T}$ (supra-threshold regime) corresponds to the case of an “oscillator”, when the discharge generation takes place even without noise. On the contrary, a positive $V^{T}$ (subthreshold regime) corresponds to the case of a “stochastic generator” with the frequency being equal if there is no noise.

With increasing stimulus amplitude $I_{stim}$, the sensitivity function becomes steeper and shifts to the left (Figure 2D, next to the top). The increase in the noise $\sigma_{V}$ leads to a shorter $T_{0}$ and a smoother $P_{con}(T)$ (Figure 2C, middle); the sensitivity function flattens (Figure 2D, middle). The increase in membrane conductance $g_{L}$ shortens $T_{0}$, due to shortening the membrane time constant, and sharpens $P_{con}$ (Figure 2C, next to the bottom); the sensitivity function shifts to the left (Figure 2D, next to the bottom). The elevation of the reset level $V_{reset}$ increases $T_{0}$ and shifts but does not significantly change the shape of $P_{con}$ for $T_{0}$ big enough in comparison to the membrane time constant, i.e., for $V^{T}\gg V_{reset}$ (Figure 2C, bottom), the sensitivity function flattens and shifts to the left (Figure 2D, bottom). We then compared the sensitivity functions and the relative ISI distributions, obtained with different parameter sets (Figure 2C,D and Figure 4A), to the experimental observations shown in Figure 1. We solved an optimization problem of fitting four model parameters, ${\sigma_{V},\;V}^{T},\;I_{stim}$, and $V_{reset}$, to minimize the residual error of four evaluated values: the mean sensitivity for three phase values, corresponding to Figure 1C, and the mean CV of the control ISI distribution, given in Table 2 (see Section 4.2.6 “Optimization Problem for Parameter Fitting” in Methods). The optimized set of parameters is given in the caption of Figure 4. It provides the sensitivity function shown in Figure 4A with the red line. The negative value of the threshold indicates that the superthreshold LIF model provides the best match to the experimental data. This case corresponds to the case of an “oscillator”.

#### 2.4.8. LIF Model: Normalization of ISI 

For the analysis described above, we used the distribution of the ISI itself or its ratio to the mean value $\langle T_{con}\rangle .$ In the experiments, in order to diminish the effects of slow processes during the period of recordings, we scaled the ISIs by the previous interval without stimulation, $T_{con}$. Does it make a difference? Figure 3B shows that the difference is not essential.

#### 2.4.9. LIF Model with M-Channels: Class I and Class II

Considering models alternative to LIF, we now study the effect of the class of excitability, which crucially changes the dependence of the firing rate on stimulus amplitude [15] and thus may affect the sensitivity function. It is known from studies of neuron models that the class of excitability changes after the addition of slow potassium M-channels [16]. We have added the M-channels to the LIF model. The approximation for the M-channels was taken from [1], and the time scale of the M-channel kinetics was rescaled with the ratio $\langle T_{con}\rangle /10\;ms$ (see Methods).

For the modified model, the RD approach is supplied with the equation derived from Equation (30) of the Methods section:(20)$$\frac{dw}{dt^{*}}=\frac{w_{\infty }(U)-w}{\tau_{w}(U)},w(t^{*}=0)=w_{\infty }(0)$$

Equation (13) is modified as follows:(21)$$C\frac{dU}{dt^{*}}=-g_{L}U-g_{KM}w\left(t\right)U+I,\;U\left(t^{*}=0\right)=V_{reset}$$

Now, the equations for the calculation of P(T) are (6)–(10), (20), and (21).

The resulting f-I curve obtained with $g_{KM}=3\;nS$ and $I_{ext}=4\;pA$ is compared in Figure 4B with that for the LIF model with the control set of parameters. The f-I curve of the stochastic LIF+KM model is S-shaped, whereas the steep fragment corresponds to the break in the deterministic model. The sensitivity function for the LIF+KM model is steeper than for the LIF model (Figure 4C, blue); it is also less sensitive to the change in the voltage threshold (the blue and green lines are close to each other). The ISI distribution of the LIF+KM model in the subthreshold case of $V^{T}=2\;mV$ (Figure 4D) shows a CV comparable with experimental values (CV = 0.26, Table 2); however, the probability of discharges drops inconsistently ($P_{next}=0.015)$. These observations indicate that the Class I model better matches the experiment than the Class II model.

#### 2.4.10. LIF Model: Direct Numerical Simulations

For direct numerical simulations with the LIF model, we used the optimal set of parameters obtained from a previous analysis performed with the help of the RD approach. The simulations are shown in Figure 5. The simulated traces (Figure 1A) do not resemble the recordings (Figure 1) because the potential in the LIF model is not supposed to be compared to the registered LFP; however, the statistics of the events and the ISI distributions (Figure 5B) are comparable. By fitting the obtained distributions (Figure 5B, right panels) with Equation (3), we obtained the sensitivity γ as a function of the phase of stimulation φ, which matches the experimental data (Figure 5C).

#### 2.4.11. Model 1: Fast Subsystem of Epileptor-2 Model with Synaptic Resource

We simulated the regime of short burst generation with the subsystem of the Epileptor-2 model that takes into account only fast processes and neglects the slow ionic dynamics. It is expressed by the stochastic ordinary differential system of the second order (see Methods). As shown in [8], the model produces short bursts as large-amplitude stochastic oscillations, and it does not show any oscillations in the absence of noise. Due to the stochastic nature of oscillations, the dispersion of interdischarge intervals is large for all the parameter sets that we tried.

One of the solutions obtained in the subthreshold conditions with $V^{T}=25\;mV$ is shown in Figure 6A. A discharge occurs spontaneously when the potential V, due to fluctuations, approaches the threshold. Then, the network excites with the firing rate ν evaluated according to the threshold sigmoid function, seen in Equation (24). The excitation results in a rapid increase in the potential and gradual exhaustion of the synaptic resource $x^{D}$ which decreases the excitation and the potential V. At the intervals of subthreshold potential V, the synaptic resource $x^{D}$ slowly restores.

The sensitivity of the system increases with the phase of stimulation (Figure 6E, red dots), as in experiment. The dependence on phase is less pronounced than in the experiment. The control interval distribution deflects from the normal distribution (Figure 6C, left), whereas the experimental distribution is close to the normal one (Figure 1D). The CV coefficient for the sub-threshold regime with $V^{T}=25\;mV$ is bigger than in experiments (0.4 in the model versus 0.2 in the experiment) due to pure stochastic nature of oscillations. The distribution of the intervals with and without stimuli (Figure 6C, right) is mono-modal, which also differs from the experiment (Figure 1B). 

The modification of the model with a smoothed sigmoidal function, Equation (24a), allows us to perform a bifurcation analysis. In Model 1, with the smooth sigmoidal function, Equation (24a), and without noise, the deterministic oscillations emerge with decreasing $V^{T}$ (or increasing $I_{ext}$) via SNIC bifurcation (Figure 6F). Therefore, the oscillations start rising in their frequency from zero, whereas their amplitude is large. No bi-stability is observed for this model; either a stable limit cycle (green) or stable fixed-point solution is observed. We observe deterministic oscillations (Figure 6F, green) in supra-threshold conditions in some ranges of $V^{T}$. With the value from that range, $V^{T}=17\;mV$, and with additional noise, the model shows oscillations with the desired CV = 0.2 (Figure 6B,D); however, the sensitivity function is too steep in this regime (Figure 6E, green dots). So, though the voltage traces simulated with Model 1 are much more comparable with the experiments than those for the LIF model, due to the observed pauses between the discharges in the former model, the characteristics of its activity in both considered regimes do not show a proper match to the experiments.

#### 2.4.12. Model 2: Fast Subsystem of Epileptor-2 Model with Shunting

As an alternative to synaptic resource exhaustion as the dominating process that terminates each short burst, in this section, we consider the calcium-dependent potassium channels which are known to be able to modulate short epileptiform discharges [17]. Because the calcium entry is mostly triggered by spikes, and because the potassium reversal potential is close to the resting potential, these channels can be approximated as slow spike-triggered shunting channels. Accordingly, we modify the equations of Model 1 by assuming a constant $x^{D}$ and instead introduce the changing conductance $G_{a}(t)$ which is driven by the firing rate and relaxes with the time constant $\tau_{a}$ according to Equation (26).

Depending on the parameters, Model 2 shows either pure stochastic oscillations (for instance, with a relatively big threshold $V^{T}$, Figure 7A,C) or noisy deterministic oscillations (Figure 7B,D). In both cases, the control interval distribution is close to the normal one (Figure 7C, left), the CV coefficient is like the one in the experiment (0.21), and the distribution of the ratio of the intervals with and without stimuli is bi-modal for early phases φ=0.3 and 0.5, which is also consistent with the experiment (Figure 1D). The sensitivity functions are close to the experimental ones in both cases (Figure 1E).

Considering the bifurcation diagram for Model 2 without noise (Figure 7F), we observe the oscillations that emerge with the decreasing $V^{T}$ (or increasing $I_{ext}$) via subcritical Hopf bifurcation followed by the saddle-node bifurcation of cycles, occurring in very narrow regions of $V^{T}$ about $V^{T}$ = 13 mV (Figure 7F). Therefore, contrary to Model 1, the oscillations start from finite frequency, whereas their amplitude is large, similar to Model 1. In the presence of noise, the starting frequency is zero in both models. Also, no bi-stability region is observed for both models.

#### 2.4.13. Model 2 with Shunting and After-Spike Depolarization

A qualitatively different sensitivity function is observed in the extended Model 2 with an additional equation for after-spike depolarization (Figure 8). Due to the after-spike depolarizing current, and in spite of strong shunting, the potential at the early phases after a discharge is depolarized and is closer to the threshold than in the later phase before the next discharge (Figure 8A). This effect provides higher sensitivity to stimuli (Figure 8C). In the experimental range of the phases of stimulation (0.3–0.7), the dependence γ(φ) is decreasing, contrary to the increasing experimental function. The ISI distributions and ISI ratios are also quite different, not showing a Gaussian-like shape for the control ISI distributions (Figure 8B, left), and not showing two distinct components in the distribution of the ratios of intervals with and without stimuli (Figure 8B, right).

## 3. Discussion

*Experiments and sensitivity function.* In one of the canonical experimental models of epileptic activity, we have studied the status epilepticus regime of activity with spontaneously repeating short discharges. We focused on the effects of weak extracellular stimulation that slightly changes the statistics of the discharges. We applied a closed-loop stimulation paradigm in order to affect the discharge generation in a certain phase between the discharges. Our experiments reveal a few characteristics of burst generation in status epilepticus. Namely, we measured four values: the coefficient of the ISI variation and the three values of sensitivity to the stimulus at three different phases of stimulus onset between the discharges. The procedure of the sensitivity measurements is our first important result. It can be used in other studies of burst generation and stimulation that elicits all-or-none responses. To our knowledge, such a sensitivity function has not been considered before.

Supposing that the shape of the introduced sensitivity function reflects the nature of the network as a pulse generator, we further analyzed this function in mathematical models. We hypothesized the sensitivity function to be different for the pure stochastic oscillators that are unable to generate spikes without noise, and for the oscillators for which the noise just perturbates the oscillations. We have failed to find a qualitative difference in the form of the sensitivity function in these two cases; however, we have seen that the sensitivity function can be one of the important criteria to distinguish between the models.

The observed interictal discharges are populational synchronization events involving a large number of neurons in the network with a complex internal structure. It is interesting that at this scale of observation, i.e., point LFP recordings, the effect of stimulation on a complex oscillating network is well described by the elementary point LIF model of neuron excitation. To some extent, we consider the dynamic of the underlying processes in an entire complex network, through a simple point neuron model. From this point, we can emphasize observable characteristic properties of a complex system by drawing analogies to the phenomena of point excitation models, such as the threshold nature of excitation that triggers emerging discharges or network-averaged resource accumulation.

*Interictal-like discharge generation regime.* Various types of epileptiform synchronization can be observed when slices are maintained in vitro and continuously perfused with a solution containing convulsant drugs and/or low [Mg^2+^] [18]. We have analyzed the 4-aminopyridine-based model of epileptiform activity, specifically focusing on the regime of the quasi-stationary generation of short, interictal-like discharges. This type of activity resembles the continuous pattern of 4-aminopyridine-induced interictal discharges observed in [18], the status epilepticus-like activity observed in [4] with the 0-Mg experimental model, and the combined 4-aminopyridine and low-Mg-based model [19], despite their different characteristics [20]. Although a detailed study of the mechanisms of the discharges is beyond the scope of the present work, and the results cannot be directly extended to other experimental models, our method of LFP-based analysis of the ISI statistics and sensitivity function is applicable to a wide range of quasi-stationary patterns of discharge generation undergoing phase-dependent stimulation. Particularly, in future studies, the sensitivity measurements can be performed in in vivo applications with the electrical registration and optogenetic stimulation of a brain area being a focus of epileptic activity. The phase dependence of the light pulses could optimize the impact of stimulation and help to avoid the problem of opsin inactivation occurring when light is prolonged.

*RD approach for ISI distributions*. The sensitivity function is based on ISI distributions. In order to perform a parametric analysis of the sensitivity function, the calculation of ISI distribution has to be computationally efficient. Unfortunately, an analytical solution for the ISI distribution is available only for particular simple models, for instance, the LIF, exponential LIF [21], or perfect integrate-and-fire model with or without adaptation [22,23]. The introduction of stimulation into models prevents using these solutions. That is why the proposed approach that reduces the integration of ordinary differential equations for a population of noise realizations to one integration of ordinary differential equations is prominent. It is applicable to a rather wide range of multi-dimensional threshold-type models; however, we are applying it here only to the LIF model with stimulation.

*LIF model.* As the simplest model of burst generation, we have considered a model that is similar to the LIF model for a neuron. We have shown that, in this simple case, the sensitivity function and the ISI distributions can be calculated semi-analytically with the help of the refractory density approach. This semi-analytical approach in application to ISI distributions is novel. The RD approach with the hazard function has been found useful and computationally effective for the calculation of ISI distributions and the sensitivity function derived from them. This approach has allowed us to perform the parametric analysis of the sensitivity function and to solve the optimization problem of fitting parameters to experimental data. To our knowledge, the RD approach has not been applied before for the statistical analysis of stimulus-evoked events.

The obtained results of optimization have shown that the LIF model matches data in both sub- and superthreshold regimes, with a precision comparable with the data dispersion; however, the best fit corresponds to the superthreshold oscillator regime. Though the LIF model is a minimal mathematical model that reproduces experimental data, it does not reflect the biophysical mechanisms of the potential relaxation between discharges and their resetting.

*Model 1*. Our first biophysical model is the fast two-dimensional subsystem of the Epileptor-2 model [8], which reproduces interictal-like discharges and assumes that the major process of each burst termination is the exhausting of synaptic resources. We however found that this model is characterized by either too large a dispersion of ISIs or too steep a sensitivity function.

*Model 2*. An alternative biophysical model explains the termination of each of the discharges by the shunting effect of slow spike-triggered conductances of presumably potassium calcium-dependent channels, as in our detailed refractory-density-based model of interictal discharges [10,24]. This model provides a relatively accurate match to the experimental data in two regimes, with more precision for the “oscillator”, which is also consistent with the previous parametric analysis of the LIF model. Bifurcation analysis has revealed different scenarios of entering the cycling for the deterministic versions of Models 1 and 2. One feature that is different is whether the oscillations start from a zero or non-zero frequency near bifurcations; it can help to distinguish the models. Model 2 always shows non-zero frequency, which is also more consistent with the all-or-none character of the discharging regimes observed in our experiment.

*Classes of excitability.* We have checked whether the sensitivity function depends on the class of excitability of the modeled network. In the absence of noise, Class I corresponds to a continuous firing rate-versus-current dependence, and Class II performs that with a jump from 0 to finite frequency. We considered the LIF model with KM-channels that belong to Class II. In the presence of noise, the sensitivity function is increasing for both classes but is steeper for Class II.

*“Wrong” model*. A qualitatively different profile of the sensitivity function has been found for the extended Model 2. It was extended to the three-dimensional system that includes a short-term, rate-dependent depolarizing current. In this case, the sensitivity function is rather flat, with a decreasing phase. This result indicates that the observation of the sensitivity function as an increasing one is a non-trivial result.

Altogether, our results reveal the best consistency between models and experiments for Model 2, based on the comparison of the interdischarge interval statistics and the sensitivity function. However, we admit that the extracellular registrations used in the present study do not provide a sufficient dataset to robustly distinguish between the two biophysical mechanisms contributing to the generation of discharges in the status epilepticus regime, including the considered processes of the synaptic resource exhaustion (Model 1) and the shunting effect of slow potassium channels (Model 2). Nevertheless, first, the study justifies in favor of the shunting-based mechanism. And second, the developed method of analysis, based on the sensitivity function and the RD approach application to ISI distributions, can be applied for the consideration of alternative event-triggering processes with stimulation. Overall, our study contributes to the development of closed-loop technology for brain stimulation.

## 4. Methods

### 4.1. Experimental Techniques

#### 4.1.1. Electrophysiological Recordings 

Wistar rats of both sexes aged 21–23 days (*n* = 8) were used in the work. The animal experiments were conducted in accordance with the ARRIVE guidelines and were performed in accordance with the EU Directive 2010/63/EU on animal experiments. The experimental protocol was approved by the Ethics Committee of the Sechenov Institute of Evolutionary Physiology and Biochemistry (Protocol No. 1-7/2022, 27 January 2022). The rats were decapitated, and the brain was quickly removed. Brain slices were prepared as previously described [25]. Briefly, horizontal brain slices that were 350 μm thick and contained the hippocampus and entorhinal cortex were sectioned in ice-cold artificial cerebrospinal fluid (ACSF) using a vibratome (HM 650V; Microm, Walldorf, Germany). The ACSF solution used in this experiment consisted of 126 mM NaCl, 24 mM NaHCO_3_, 2.5 mM KCl, 2 mM CaCl_2_, 1.25 mM NaH_2_PO_4_, 1 mM MgSO_4_, and 10 mM glucose. After cutting, the slices were then incubated at 35 °C for 1 h before electrophysiological recording.

During the experiments, the slices were perfused with ACSF at a constant flow rate of 5 mL/min. Glass microelectrodes (0.2–1.0 MΩ) were used to record extracellular field excitatory postsynaptic potentials (fEPSPs) from the CA1 stratum radiatum. Synaptic responses were elicited by the local extracellular stimulation of the Schaffer collaterals using an A365 stimulus isolator (World Precision Instruments, Sarasota, FL, USA). Local field potentials (LFPs) were recorded using a Model 1800 amplifier (A-M Systems, Carlsborg, WA, USA) and were digitized and saved using ADC/DAC NI USB-6211 (National Instruments, Austin, TX, USA) and WinWCP v5.7.8 software (University of Strathclyde, Glasgow, UK). The electrophysiological records were analyzed using the Clampfit 10.2 program (Molecular Devices, Sunnyvale, CA, USA).

The induction of epileptiform activity in the slices was carried out by the application of a proepileptic solution containing 100 µM 4-aminopyridine, 126 mM NaCl, 24 mM NaHCO_3_, 3.5 mM KCl, 2 mM CaCl_2_, 1.25 mM NaH_2_PO_4_, 0.25 mM MgSO_4_, and 10 mM glucose.

#### 4.1.2. Stimulation Protocols

The closed-loop feedback system autonomously controlled the stimulation, with the dynamic recalculation of stimulation parameters (Figure 1A). In this study, electrical stimulation was performed with a fixed current strength for each cut, and the calculated parameter was the stimulation time relative to the last interictal-like event (Figure 1B). 

The stimulus was a 0.1 ms DC step throughout all experiments, and the strength of the stimulation current was adjusted for each slice. Fifteen minutes after placing the slice in the epileptogenic solution, periodic stimulation was conducted. The current strength was increased by 5 µA every half minute until a weak population response appeared. The final value was chosen as the subthreshold value of the stimulation current, which did not cause a population response. On average, this value varied by about 35 µA.

The protocol was initiated 5 min after the onset of stable continuous periodic epileptiform activity. The recorded LFP signal was filtered to remove noise, and the emerging interactivity-like spikes were automatically detected (Figure 1B). Stimulation was performed at a specific phase (φ) relative to the duration of the previously recorded control ISI, with options of 0.3, 0.5, or 0.7. To ensure relaxation after the stimulus impact, the next interstimulus interval (ISI) was skipped regardless of the population response manifestation. If a discharge occurs earlier than the calculated phase-dependent stimulation time after the control ISI, it is considered a miss, and the corresponding interval is not accounted for, and a new control interval is considered.

#### 4.1.3. Calculation of Experimental Phase-Dependent Sensitivity to Stimulation

The experimental values of the sensitivity function were estimated by fitting the experimental $T_{stim}^{exp}/T^{exp}$ distributions by the theoretical distribution $P_{z}\left(z\right)$ defined by Equation (3) (Figure 1D,E), varying γ and δz for each value of the stimulation phase φ. For each of the φ values (0.3, 0.5, 0.7), the estimated stimulation efficiency corresponds to the optimum γ value when the experimental distribution approximates the theoretical one (Figure 1F).

### 4.2. Mathematical Methods

#### 4.2.1. Model 1: Fast Subsystem of Epileptor-2 Model with Synaptic Resource

The original Epileptor-2 model describes epileptic discharges in terms of 4 variables: spikeless membrane potentials, synaptic resources, and extracellular potassium and intracellular sodium concentrations. It reproduces interictal short discharges (SDs) as large-amplitude stochastic oscillations that are observed in a single neuron as bursts of spikes. It also reproduces ictal discharges (IDs) as clusters of SDs, determined by the oscillations of the ionic concentrations. The population model consists of ordinary differential equations written for the extracellular potassium and intracellular sodium concentrations, the spikeless membrane potential V(t), and the synaptic resource $x^{D}$. For the present study of the only regime of status epilepticus, we consider the fast subsystem of the full model, which is as follows:(22)$$C\frac{dV}{dt}=-g_{L}V+G_{syn}\nu \left(V\right)\left(x^{D}-0.5\right)+I_{ext}+\sigma \xi (t)$$
(23)$$\frac{dx^{D}}{dt}=\frac{1-x^{D}}{\tau_{D}}-\delta x^{D}x^{D}\nu (V)$$
where the firing rate ν(t) is calculated with a sigmoidal input–output function:(24)$$\nu (V)=\nu_{max}{\left[\frac{2}{1+\exp\left[-2\left(V(t)-V^{T}\right)/20\right]}-1\right]}_{+}$$

The smoothed version:(24a)$$\nu (V)=\frac{\nu_{max}}{1+\exp\left[-\left(V\left(t\right)-10-V^{T}\right)/2.4\right]}$$

The first term in Equation (1) is the leak. The second is the synaptic currents, where the excitatory current depends on the available synaptic resources, $G_{syn}\nu {\;x}^{D}$, whereas the inhibitory current does not and is assumed to be half of the maximum excitatory one, i.e., equal to $-0.5\;G_{syn}\nu$. The voltage fluctuations are determined by the Gaussian white noise ξ(t) with zero mean and unity dispersion.

The synaptic resource $x^{D}(t)$ is governed by the firing rate. It restores according to the first term in Equation (2) and decreases due to the second term which is proportional to the presynaptic firing rate.

The basic set of parameters is as follows: the time constant of synaptic resources $\tau_{D}=2\;s$; the membrane capacitance C=10 pF; the leak conductance $g_{L}=1\;nS$; the synaptic charge $G_{syn}=5\;pA\cdot s$; the noise amplitude is σ=8 pA; the maximal rate $\nu_{max}=100\;Hz$; the threshold potential $V^{T}=25\;mV$; the external current $I_{ext}=13\;pA$; and the synaptic resource decrement $\delta x^{D}=0.01$.

#### 4.2.2. Model 2: Fast Subsystem of Epileptor-2 Model with Shunting

In contrast to Equation (1), the membrane potential is changing due to not only the leak $g_{L}$ and synaptic $G_{syn}\nu$, but also shunting conductance $G_{a}$:(25)$$C\frac{dV}{dt}=-\left(g_{L}+G_{a}\right)V+G_{syn}\nu \left(t\right)+I_{ext}+\sigma \xi (t)$$
(26)$$\frac{dG_{a}}{dt}=-\frac{G_{a}}{\tau_{a}}+\delta G_{a}\nu (t)$$

The shunting conductance $G_{a}$ is the firing rate dependent with the coefficient of proportionality $\delta G_{a}$ and slowly relaxes with the time constant $\tau_{a}$. The parameters for Model 2 in the regime of “Oscillator” were C=10 pF, $g_{L}=1\;nS$, $G_{syn}=2.5\;pA\cdot s$, $V^{T}=10\;mV$, σ=11 pA, $\delta G_{a}=0.15\;nS$, $\tau_{a}=1.5\;s$, and $I_{ext}=13\;pA$. The parameters for Model 2 in the regime of “Stochastic oscillator” differed with $V^{T}=25\;mV$ and σ=17 pA.

#### 4.2.3. LIF Model

The stochastic LIF model is defined as follows:(27)$$C\frac{dV}{dt}=-g_{L}V+I_{ext}(t)+\sigma \xi (t)$$
(28)$$\mathrm{If}\;V>V^{T}\;\mathrm{then}\;V=V_{reset}$$

Here $I_{ext}(t)$ is the pulse of stimulating current applied in different phases of spiking. For the subthreshold regime, we set $V^{T}>0$, and $V^{T}<0$ for the superthreshold regime.

The control parameter values were $V_{reset}=-20\mathrm{mV}$, $V^{T}=-1\mathrm{mV}$ (superthreshold regime), $\sigma_{V}=1\mathrm{mV}$, $g_{L}=1\mathrm{nS}$, C=1nF, $I_{stim}=10\mathrm{pA}$, ${∆t}_{stim}=200\mathrm{ms}$, and $I_{ext}=0$.

#### 4.2.4. LIF Model with M-Channels

The M-channels are added to the LIF model. The approximation of M-channels is taken from [1] and modified with the voltage shift of $\left({-65\;mV-V}_{reset}\right)$ and the time rescaling of 100 times, thus obtaining
(29)$$C\frac{dV}{dt}=-g_{L}V-g_{KM}w\left(t\right)V+I_{ext}(t)+\sigma \xi (t)$$
(30)$$\frac{dw}{dt}=\frac{w_{\infty }-w}{\tau_{w}},$$
with $w_{\infty }=\frac{1}{1+\exp((21-V+V_{reset})/8)}$, $\tau_{w}=2000+32000\;exp\left(-\frac{{\left(15-V+V_{reset}\right)}^{2}}{625}\right)$
(31)$$\mathrm{If}\;V>V^{T},\;\mathrm{then}\;V=V_{reset},\;\mathrm{and}\;w=w_{\infty }(0).$$

Because of the shift of the f-I curve, for the case with $g_{KM}=3\;nS$, we used $I_{ext}=4\;pA$.

#### 4.2.5. Model 2 with Shunting and After-Spike Depolarization

The model is an extension of the Model 2 to the three-dimensional case that introduces the spike-evoked depolarizing current $I_{a}$:(32)$$C\frac{dV}{dt}=-\left(g_{L}+G_{a}\right)V+G_{syn}\nu \left(t\right)+I_{ext}+I_{a}+\sigma \xi (t)$$
(33)$$\frac{dG_{a}}{dt}=-\frac{G_{a}}{\tau_{a}}+\delta G_{a}\nu (t)$$
(34)$$\frac{dI_{a}}{dt}=-\frac{I_{a}}{\tau_{i}}+\delta I_{a}\nu (t)$$

The after-spike depolarizing current $I_{a}$ is the firing rate dependent on the coefficient $\delta I_{a}$ and slowly relaxed with the time constant $\tau_{i}$. The parameters for Model 2 were C=10 pF, $g_{L}=1\;nS$, $G_{syn}=2.5\;pA\cdot s$, $V^{T}=25\;mV$, σ=13 pA, $\delta G_{a}=0.15\;nS$, $\tau_{a}=1.5\;s$, $I_{ext}=13\;pA$, $\delta I_{a}=4\;pA\times ms$, and $\tau_{i}=1\;s$.

#### 4.2.6. Optimization Problem for Parameter Fitting

The parameters of the LIF model, ${\sigma_{V},V}^{T},I_{stim}$, and $V_{reset}$, have been fitted to the four experimental measurements: the three mean sensitivity values (0.034, 0.14, and 0.36) obtained for the three phases (0.3, 0.5, and 0.7), corresponding to Figure 1D, and the mean CV of the control ISI distribution (0.2) given in Table 2. Four model parameters have been fitted, ${\sigma_{V},\;V}^{T},\;I_{stim}$, and $V_{reset}$. The target function has been set in the following form:(35)$$\underset{{\sigma_{V},V}^{T},I_{\mathrm{stim}},V_{\mathrm{reset}}}{\min}\left\{100{\left(0.034-\gamma \left(0.3\right)\right)}^{2}+{\left(0.14-\gamma \left(0.5\right)\right)}^{2}+{\left(0.36-\gamma \left(0.7\right)\right)}^{2}+10{\left(0.2-CV\right)}^{2}\right\}$$

The obtained parameters are given in the figure captions.

## Figures and Tables

**Figure 1 ijms-25-08287-f001:**
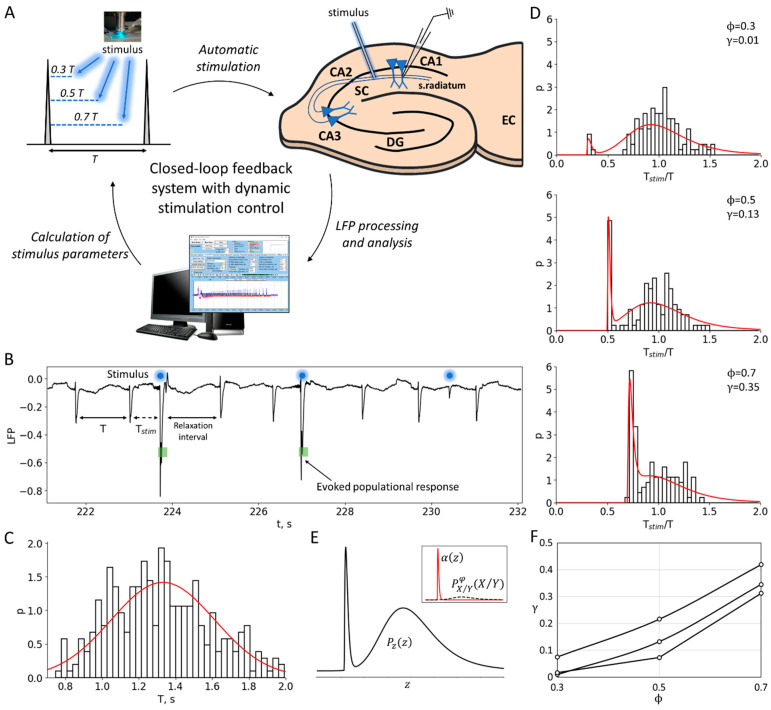
Experiment and data processing. (**A**) The closed-loop feedback system with a dynamic control of the stimulation. The system uses software to process and analyze LFP signals derived from the CA1 region of rat hippocampal slices in real time. The temporal characteristics of interictal-like population discharges during status epilepticus are used to calculate the parameters for subsequent stimulation. (**B**) Experimental LFP recording during status epilepticus. Stimulation is applied at a certain phase between the discharges φ relative to the previous control interspike interval (ISI), i.e., at the time moment t=φ T after the control interval. The population response has a probabilistic nature and depends on the phase of stimulation. (**C**) An example of the control ISI distribution during status epilepticus, $P_{con}(T)$. (**D**) The distributions of the ratio of the intervals with and without stimulation, $P_{z}\left(z\right)=P_{z}(T_{stim}/T)$, for three different phases φ (0.3, 0.5 and 0.7). The sharp peak in each distribution corresponds to the evoked responses in a certain phase φ. The red curves show theoretical distributions $P_{z}\left(z\right)$, taking into account the estimated value of the stimulation efficiency γ. (**E**) The theoretical ratio distribution $P_{z}\left(z\right)$ is calculated as the sum of $\alpha \left(z\right)$ and $P_{X/Y}^{\varphi }(z)$, based on the estimated value of the stimulation efficiency (see details in the Section 4). $P_{X/Y}^{\varphi }(z)$ corresponds to the ISI ratio distribution in the absence of stimulation, where z is the X/Y ratio, and X and Y are both distributed according to $P_{con}$. The evoked component of $P_{z}\left(z\right)$ is approximated by the shape $\alpha \left(z\right)$ (see Section 2.2 “Phase-Dependent Sensitivity to Stimulation”). (**F**) Stimulation efficiency γ curves estimated for three sets of experimental data.

**Figure 2 ijms-25-08287-f002:**
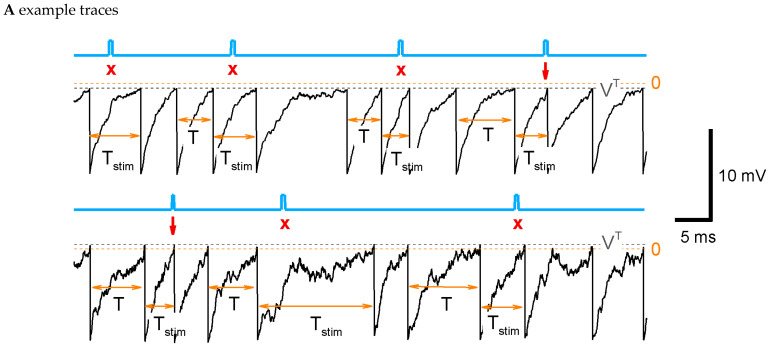
LIF model: ISI distribution, phase-dependent sensitivity function, and its dependence on parameters. (**A**) Example traces for the control parameter set of a supra-threshold regime (top) with {$V_{reset}=-20\;mV$, $V^{T}=-1\;mV$, $\sigma_{V}=1\;mV$, $g_{L}=1\;nS$, C=1 nF, $I_{stim}=10\;pA$, ${∆t}_{stim}=200\;ms$, $I_{ext}=0$} and the subthreshold regime (bottom) with $V^{T}=1\;mV$, $\sigma_{V}=2.4\;mV$. The blue line marks the stimulus; the gray dashed line is the threshold; and the orange dashed line is the zero level. The stimulation is applied at φ=0.5, i.e., at $t=0.5\;T_{0}$, where $T_{0}$ is the mean control interval. Arrows mark successive stimuli, and crosses mark unsuccessful stimuli. (**B**) The steps of the calculation of sensitivity to the stimulus for the LIF model. Top to bottom, for two cases with (red) and without stimulus (black) at the phase $\varphi =t^{*}/T_{0}=0.5$: the membrane potential versus time since the previous spike; the hazard of new spike generation H; the density ρ; the ISI distribution; the distribution of the ratio of ISI after stimulation versus control ISI, cut at φ=0.5 (red), and its approximation with Equation (3) (green). (**C**,**D**) The distribution of ISI (**C**) and the phase dependence of the sensitivity γ (**D**) calculated with the control parameter values and with modified values for one of the parameters (top to bottom): the voltage threshold $V^{T}$; the stimulation current $I_{stim}$; the noise amplitude $\sigma_{V}$; the leak conductance $g_{L}$; and the reset potential $V_{reset}$. Legends for (**C**,**D**) are the same. The control case is marked by the black line.

**Figure 3 ijms-25-08287-f003:**
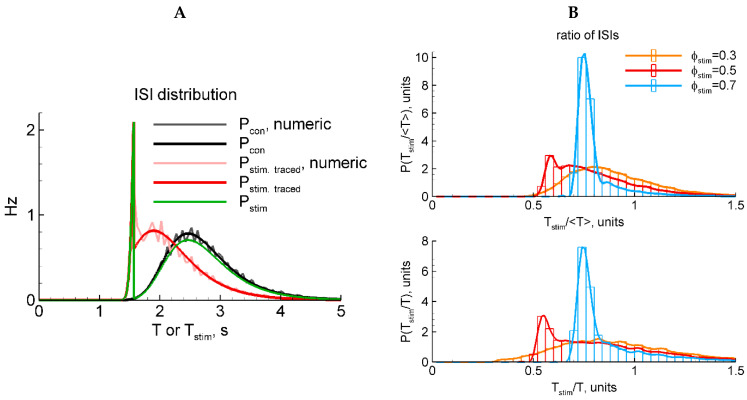
LIF model: two types of stimulation (**A**) and two types of ISI normalization (**B**). (**A**) Black and gray, analytical and numerical solutions, respectively, for $P_{con}(T)$ in the control case without stimulation. Red and rose, analytical and numerical solutions, respectively, for $P_{stim.\;traced}(T)$ in the case of stimulation that is traced in the membrane potential after the end of the stimulus. Green, $P_{stim}(T)$ in the case of stimulation that does not affect the membrane potential after the stimulus. (**B**) Numerically calculated distributions for the ratio of the interval with the evoked discharge to either the mean control ISI (top panel) or the previous control ISI (bottom). The red distribution ${P(T}_{stim}/\langle T\rangle )$ in (**B**) corresponds to the rescaled distribution of $P_{stim.\;traced}(T)$ in (**A**).

**Figure 4 ijms-25-08287-f004:**
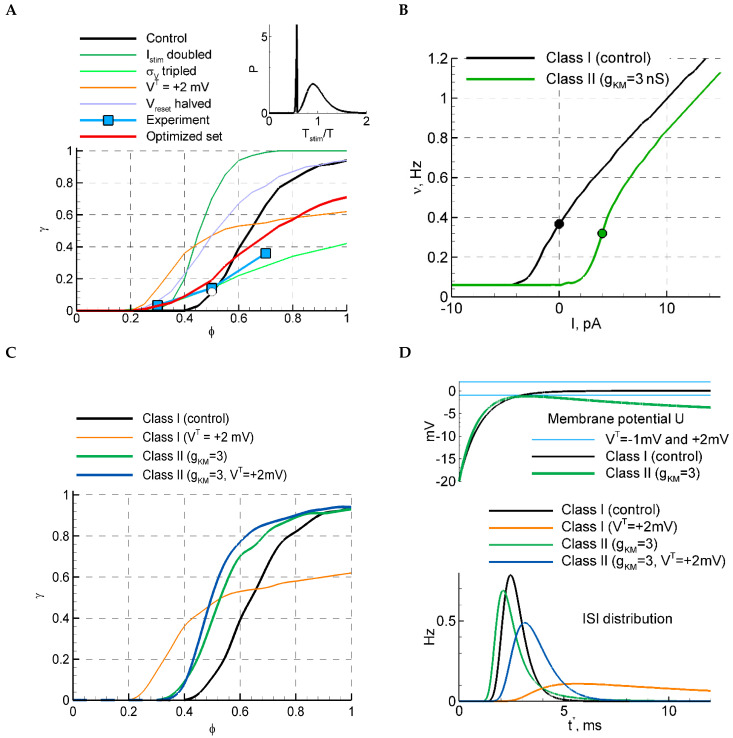
LIF model with and without M-channels. (**A**) The selection of parameters based on the sensitivity function and the relative ISI distribution (inset) for the LIF model. For the red curve, the following three parameters were fitted: $\sigma_{V}=3.3\;mV$, $V^{T}=-8.8\;mV$, $I_{stim}=16\;pA$, and $V_{reset}=-32\;mV$, and the rest of the parameters are from the control set; the other curves are replotted from Figure 2D. The ISI distribution in the inset corresponds to the white circle at φ=0.5 on the black curve. The blue squares correspond to the mean over experimental data points shown in Figure 1C. (**B**) f-I curves for models of different classes of excitability, the control LIF model (Class I), and the LIF model with M-channels (Class II). (**C**) The sensitivity function for the two models with $V^{T}=-1\;mV$ and $V^{T}=2\;mV$. In the case of $V^{T}=-1\;mV$, the firing rates in LIF and LIF+KM models correspond to the values from the black and green dots in (**B**), respectively. (**D**) ISI distributions (bottom) and the membrane potential evolution after the discharge at $t^{*}=0$, for LIF and LIF+KM models.

**Figure 5 ijms-25-08287-f005:**
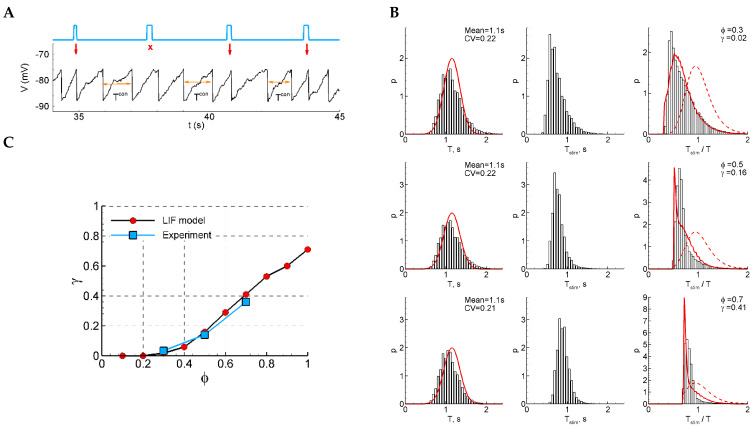
LIF model. (**A**) An example of a single trace with three light pulse stimuli, successful (red arrow) and unsuccessful (crosses). (**B**) The distributions of the control interval (left column), the interval with a stimulus (middle), and the ratio of the intervals (right) for different phases of stimulation (φ=0.3,0.5 and 0.7). The red curves are the Gaussian profile fitted to the control interval distribution (left) and the $P_{z}\left(z\right)$ approximation (right). (**C**) The sensitivity function in comparison with the mean experimental data. The model parameters were $V_{reset}=-18\mathrm{mV}$, $V^{T}=-5.6\mathrm{mV}$, $\sigma_{V}=1.74\mathrm{mV}$, $g_{L}=1\mathrm{nS}$, C=1nF, $I_{stim}=20\mathrm{pA}$, ${∆t}_{stim}=200\mathrm{ms}$, and $I_{ext}=0$.

**Figure 6 ijms-25-08287-f006:**
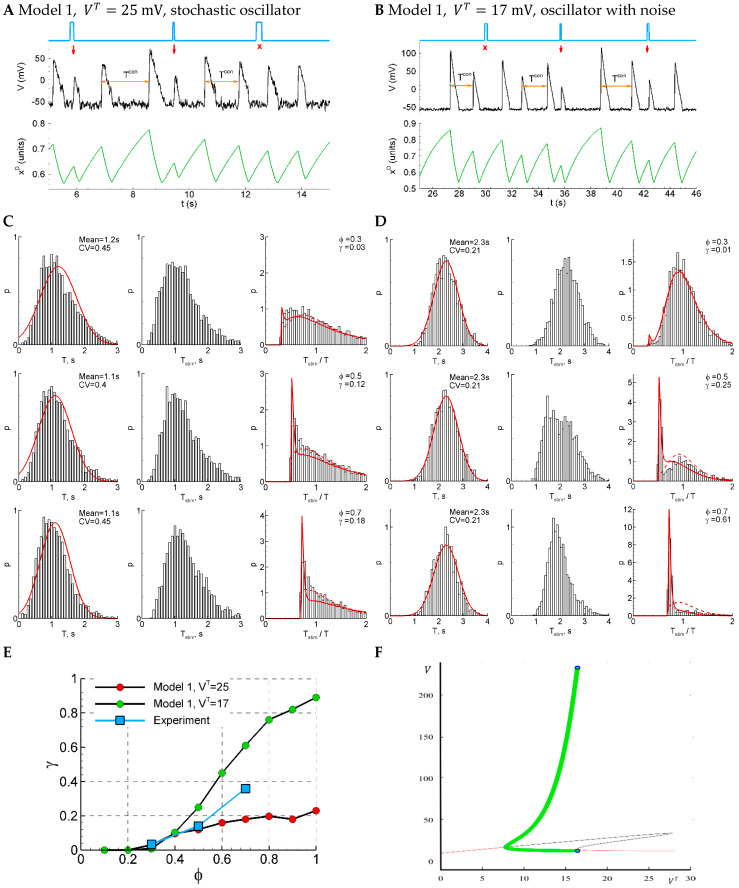
Model 1 is similar to the experiment by either sensitivity function or CV, but not both of them, comparing solutions in subthreshold ($V^{T}=25\;mV$, panels (**A**,**C**)) and supra-threshold ($V^{T}=17\;mV$, panels (**B**,**D**)) regimes. (**A**,**B**) An example of a single trace with three light pulse stimuli, successful (red arrows) and unsuccessful (cross). (**C**,**D**) The distributions of the control interval (left column), the interval with the stimulus (middle), and the ratio of the intervals (right) for different phases of stimulation (φ=0.3, 0.5, and 0.7). The red curves are the Gaussian profile fitted to the control interval distribution and the $P_{z}\left(z\right)$ approximation. (**E**) The sensitivity function in comparison with the mean experimental data. (**F**) A bifurcation diagram for Model 1 without noise. The cycle (green) appears with the decreasing $V^{T}$ at about $V^{T}=17\;mV$ via SNIC bifurcation; it disappears at about $V^{T}=8\;mV$ via supercritical Hopf bifurcation. The red lines correspond to stable fixed-point solutions.

**Figure 7 ijms-25-08287-f007:**
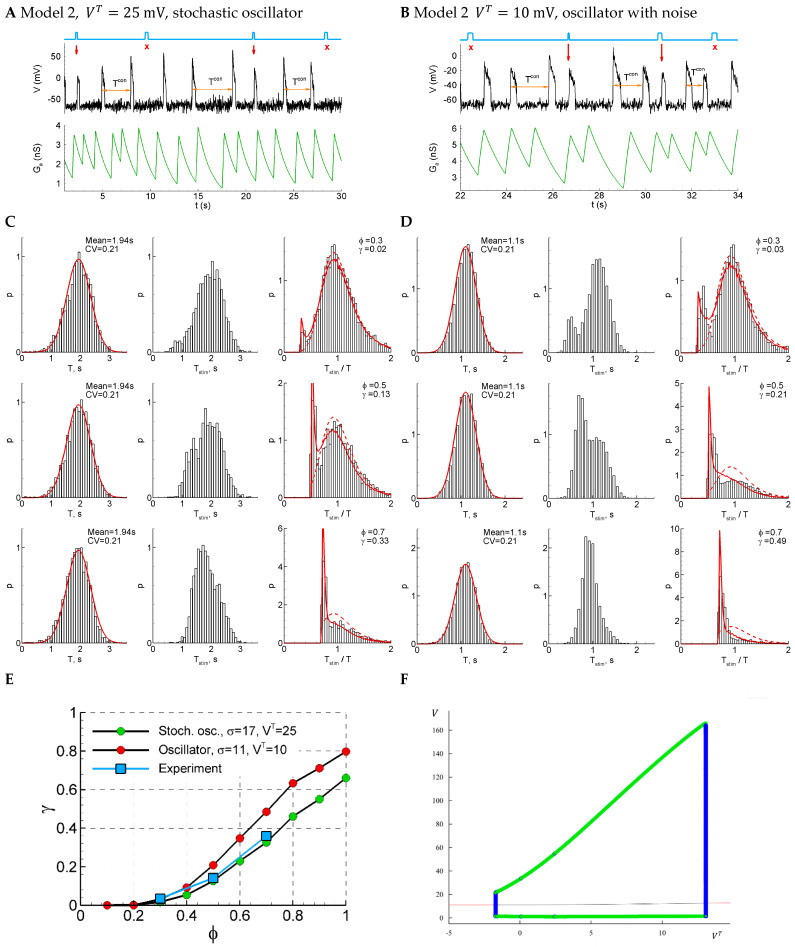
Model 2 in two regimes, subthreshold ($V^{T}=25\;mV$, panels (**A**,**C**)) and supra-threshold ($V^{T}=10\;mV$, panels (**B**,**D**)), in comparison with the experiment. (**A**,**B**) An example of a single trace with three light pulse stimuli, successful (red arrows) and unsuccessful (cross). (**C**,**D**) The distributions of the control interval (left column), the interval with the stimulus (middle), and the ratio of the intervals (right) for different phases of stimulation (φ=0.3, 0.5, and 0.7). The red curves are the Gaussian profile fitted to the control interval distribution and the $P_{z}\left(z\right)$ approximation. (**E**) The sensitivity functions in comparison with the mean experimental data. (**F**) The bifurcation diagram for Model 2 without noise and a smooth sigmoid function, as in Equation (24a). The cycle appears and disappears via subcritical Hopf bifurcations followed by the saddle-node of the cycles.

**Figure 8 ijms-25-08287-f008:**
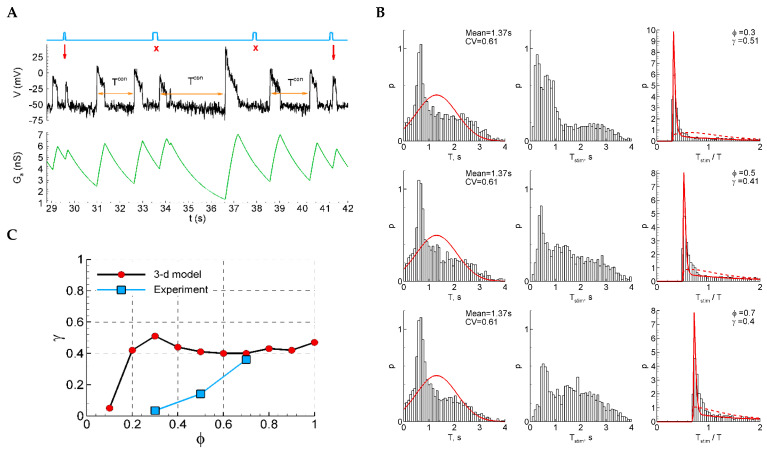
Model 2 with after-spike depolarization (“3-D model”). (**A**) An example of a single trace with three light pulse stimuli, successful (red arrows) and unsuccessful (cross). (**B**) The distributions of the control interval (left column), the interval with the stimulus (middle), and the ratio of the intervals (right) for different phases of stimulation (φ=0.3, 0.5 and 0.7). The red curves are the Gaussian profile fitted to the control interval distribution and the $P_{z}\left(z\right)$ approximation. (**C**) The sensitivity function in comparison with the mean experimental data.

**Table 1 ijms-25-08287-t001:** List of models.

	Models	Type	Order of ODEs	Variables
1	Fast subsystem of Epileptor-2 with synaptic resource	Stochastic, mean-field based on sigmoid function	2	membrane potential V and synaptic resource $x^{D}$
2	Fast subsystem of Epileptor-2 with shunting conductance	Stochastic, mean-field based on sigmoid function	2	membrane potential V and shunting conductance $G_{a}$
3	LIF model	Stochastic, threshold	1	membrane potential V
4	LIF model with KM-channel	Stochastic, threshold, with HH-like approximation	2	membrane potential V and M-channel conductance w
5	Model 2 with after-spike depolarization	Stochastic, mean-field based on sigmoid function	3	membrane potential V, shunting conductance $G_{a}$ and after-spike depolarizing current $I_{a}$

**Table 2 ijms-25-08287-t002:** The mean and the coefficient of ISI variation (CV) for experiments and models. The control case corresponds to the parameter set given in the caption of Figure 2.

Model/Experiment	Parameters Differed from the Control Set	Mean IS, s	CV	$P_{next}$.
Experiments	-	1.49 (1.54, 1.60, 1.34)	0.20 (0.15, 0.21, 0.23)	1
LIF	Control	2.72	0.22	1
LIF	$V^{T}=0$	3.65	0.30	1
LIF	$V^{T}=+1\;mV$	5.72	0.43	1
LIF	$V^{T}=+2mV$	8.76	0.57	1
LIF	$\sigma_{V}=3\;mV$	2.18	0.41	1
LIF	$\sigma_{V}=0.5\;mV$	2.88	0.14	1
LIF	$g_{L}=2\;nS$	1.36	0.22	1
LIF	$g_{L}=0.5\;nS$	5.46	0.22	1
LIF	$V_{reset}=-3\;mV$	0.87	0.67	1
LIF	$V_{reset}=-40\;mV$	3.41	0.18	1
LIF+KM	$g_{KM}=3\;nS$	2.76	0.45	1
LIF+KM	$g_{KM}=3\;nS$ $,\;V^{T}=2\;mV$	3.62	0.26	0.015
Model 1 with syn. resource, “Stoch. oscillator”	$V^{T}=25\;mV$ , σ=8 pA	1.1	0.45	1
Model 1 with syn. resource, close to “Oscillator”	$V^{T}=17\;mV$ , σ=2.5 pA	2.3	0.21	1
Model 2 with shunting, “Oscillator”	$V^{T}=10\;mV$ , σ=11 pA	1.9	0.21	1
Model 2 with shunting, “Stoch. oscillator”	$V^{T}=25\;mV$ , σ=17 pA	1.1	0.21	1

## Data Availability

The data presented in this study are available upon request from the corresponding author.

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
