# Peer review of "Phase-Dependent Response to Electrical Stimulation of Cortical Networks during Recurrent Epileptiform Short Discharge Generation In Vitro"

_ijms, 2024, doi:10.3390/ijms25158287_

Round 1

Reviewer 1 Report

Comments and Suggestions for Authors

This manuscript presents a comprehensive investigation into the sensitivity of continuous epileptiform activity to electrical stimulation. The authors evaluate a series of biophysical and mathematical models to characterize this sensitivity. Their methodology involves applying electrical pulses at various intervals between epileptiform discharges in an in vitro model of epilepsy, utilizing rat hippocampal slices treated with 4-aminopyridine.

To quantify the effectiveness of stimulation, the researchers introduce a sensitivity function, which they subsequently analyze using different models. Their findings suggest that the Epileptor-2B model (Model 2) is the most suitable for describing subthreshold oscillations in the presence of significant noise. This conclusion is drawn from a comparison of inter-spike interval (ISI) statistics and sensitivity functions between the model predictions and experimental data.

The study emphasizes two key points: the stochastic nature of epileptiform discharge generation and the enhanced efficacy of closed-loop stimulation during the later phases of ISIs. The authors present their results in a well-organized manner, providing a thorough analysis of various models and their capacity to replicate experimental observations.

Overall, this manuscript offers a detailed and rigorous examination of the relationship between electrical stimulation and epileptiform activity, contributing valuable insights to the field of epilepsy research and potential therapeutic interventions.

Comments on the Quality of English Language

The manuscript is generally well-written, with the some areas where the quality of the English language can be improved for clarity, coherence, and readability. However it is not critical.

Author Response

Comment: "The manuscript is generally well-written, with the some areas where the quality of the English language can be improved for clarity, coherence, and readability. However it is not critical."

Response: We have introduced minor corrections of English grammar throughout the text.

Reviewer 2 Report

Comments and Suggestions for Authors

This manuscript demonstrates a sensitivity function to measure stimulation effectiveness in experiments and computational models. This analytical method is applied by the authors to examine several mathematical models for their performances in describing the physiological epileptic network activity in the rat hippocampus. The concept is very interesting and can be useful in examining the performance of other neurocomputational models. My comments are primarily minor, as listed below:

  1. Please point out a few future directions for this sensitivity measurement in in vivo applications.
  2. Specify the details of the animals (e.g., sex, age, etc.) and include a statement that the animal experimental procedures followed the approval of the facility.
Comments on the Quality of English Language

Good enough.

Author Response

Comment1: "Please point out a few future directions for this sensitivity measurement in in vivo applications."

Response1: Thank you for this question. Of course, in vivo applications is the main direction for future studies. We have describe this point in a new paragraph of the Discussion section: 

"Interictal-like discharge generation regime. Various types of epileptiform synchronization can be observed when slices are maintained in vitro and continuously perfused with a solution containing convulsant drugs and/or low [Mg2+]o (Avoli et al., 2002). We have analyzed the 4-aminopyridine-based model of epileptiform activity, specifically focusing on the regime of quasi-stationary generation of short, interictal-like discharges. This type of activity resembles the continuous pattern of 4-aminopyridine-Induced interictal discharges observed in  [Avoli et al., 2002], the status epilepticus-like activity observed in [Burman] with the 0-Mg experimental model, and the combined 4-aminopyridine and low-Mg-based model [Scalmani et al. 2023], despite their different characteristics [Trevelyan et al. 2022]. Although a detailed study of the mechanisms of the discharges is beyond the scope of the present work, and the results cannot be directly extended to other experimental models, our method of LFP-based analysis of ISI statistics and sensitivity function is applicable to a wide range of quasi-stationary patterns of discharge generation undergoing phase-dependent stimulation. Particularly, in future studies, the sensitivity measurements can be performed in in vivo applications with electrical registration and optogenetic stimulation of a brain area being a focus of epileptic activity. The phase-dependence of the light pulses could optimize the impact of stimulation and help to avoid the problem of opsin inactivation occurring when light is prolonged."

Comment2: "Specify the details of the animals (e.g., sex, age, etc.) and include a statement that the animal experimental procedures followed the approval of the facility."

Response2: We have introduced these details in the text of the Methods section:

"Wistar rats of both sexes aged 21-23 days (n=8) were used in the work. The animal experiments were conducted in accordance with the ARRIVE guidelines and were performed in accordance with the EU Directive 2010/63/EU on animal experiments. The experimental protocol was approved by the Ethics Committee of the Sechenov Institute of Evolutionary Physiology and Biochemistry (Protocol No. 1-7/2022, 27 January 2022). "

Reviewer 3 Report

Comments and Suggestions for Authors

In this study, the sensitivity of continuous epileptiform short discharge generation to electrical stimulation was investigated using an in vitro 4-AP-based model of epilepsy in rat hippocampal slices. The authors analyzed the sensitivity of stimulation effectiveness using different biophysical and abstract mathematical models and found that sensitivity increased with phase for all models except the last one. They also emphasized the stochastic nature of epileptiform discharge generation and the greater effectiveness of closed-loop stimulation in later phases of inter-spike intervals.

After reviewing the manuscript, I find it well-written, and I don’t have any specific comments or concerns. However, I do have a general question about the potential applicability of the models to other SE models, such as 0-Mg2+. 

Additionally, I noticed that the discussion section seems to lack sufficient citations. 

Author Response

Comment1: "I do have a general question about the potential applicability of the models to other SE models, such as 0-Mg2+. "

Response1: Thank you for this question. To answer, we have introduced an additional paragraph into the Discussion section:

"Interictal-like discharge generation regime. Various types of epileptiform synchronization can be observed when slices are maintained in vitro and continuously perfused with a solution containing convulsant drugs and/or low [Mg2+]o (Avoli et al., 2002). We have analyzed the 4-aminopyridine-based model of epileptiform activity, specifically focusing on the regime of quasi-stationary generation of short, interictal-like discharges. This type of activity resembles the continuous pattern of 4-aminopyridine-Induced interictal discharges observed in  [Avoli et al., 2002], the status epilepticus-like activity observed in [Burman] with the 0-Mg experimental model, and the combined 4-aminopyridine and low-Mg-based model [Scalmani et al. 2023], despite their different characteristics [Trevelyan et al. 2022]. Although a detailed study of the mechanisms of the discharges is beyond the scope of the present work, and the results cannot be directly extended to other experimental models, our method of LFP-based analysis of ISI statistics and sensitivity function is applicable to a wide range of quasi-stationary patterns of discharge generation undergoing phase-dependent stimulation. Particularly, in future studies, the sensitivity measurements can be performed in in vivo applications with electrical registration and optogenetic stimulation of a brain area being a focus of epileptic activity. The phase-dependence of the light pulses could optimize the impact of stimulation and help to avoid the problem of opsin inactivation occurring when light is prolonged."

Comment2: "Additionally, I noticed that the discussion section seems to lack sufficient citations."

Response2: We have supported the text of Introduction and discussion with 9 additional references.